# Bioactive Compounds Discovery from French Guiana Plant Extracts Through Antitubercular Screening and Molecular Networking

**DOI:** 10.3390/plants14193028

**Published:** 2025-09-30

**Authors:** Célia Breaud, Clémentine Saunier, Béatrice Baghdikian, Fathi Mabrouki, Myriam Bertolotti, Mariana Royer, Pierre Silland, Marc Maresca, Eldar Garaev, Jean-François Cavalier, Stéphane Canaan, Sok-Siya Bun-Llopet, Elnur Garayev

**Affiliations:** 1Aix-Marseille Univ, CNRS 7263, IRD 237, Avignon Univ, IMBE, 27 bd Jean Moulin, Service of Pharmacognosy-Ethnopharmacology, Faculty of Pharmacy, 13385 Marseille, France; celia.breaud@univ-amu.fr (C.B.); beatrice.baghdikian@univ-amu.fr (B.B.); sok-siya.bun@univ-amu.fr (S.-S.B.-L.); 2Aix-Marseille Univ CNRS, LISM UMR 7255, IMM FR3479, 13009 Marseille, France; csaunier@imm.cnrs.fr (C.S.); jfcavalier@imm.cnrs.fr (J.-F.C.);; 3BioStratège SAS, 12 Lotissement Dalmazir, Z.I. Larivot, 97351 Matoury, French Guiana, France; 4Aix Marseille Univ, CNRS, Centrale Marseille, iSm2, 13013 Marseille, France; m.maresca@univ-amu.fr; 5Department of General and Toxicological Chemistry, Azerbaijan Medical University, Baku AZ1001, Azerbaijan

**Keywords:** *Mycobacterium tuberculosis*, metabolomics, flavonoids, ethnopharmacology, mass spectrometry, mycobacteria

## Abstract

Tuberculosis (TB) is still a significant public health threat, with rising drug resistance and high incidence in multiple areas worldwide. In the search for novel antitubercular agents, this study explores the application of a bioactivity-guided molecular networking approach to identify bioactive compounds from seven plant species (*Curatella americana*, *Davilla nitida*, *Dipteryx punctata*, *Indigofera suffruticosa*, *Quassia amara*, *Tetradenia riparia*, and *Zingiber zerumbet*) collected in French Guiana. Using ultrasound-assisted extraction followed by liquid–liquid partitioning and UHPLC-HRMS/MS analysis, a library of 72 samples was tested against *Mycobacterium tuberculosis*. The non-polar fractions from *Indigofera suffruticosa*, *Tetradenia riparia*, and *Zingiber zerumbet* showed the highest activity. The integration of metabolomic and bioassay data on molecular networks allowed the prioritization and annotation of active compounds, revealing flavonoids as contributors to the antitubercular activity of the active samples. In addition, the use of computational tools such as GNPS, SIRIUS, and TIMA-R enabled dereplication and increased the confidence in the structural prediction of active metabolites. This approach demonstrated its potential in accelerating the identification of both known and novel bioactive compounds without requiring exhaustive isolation, offering a robust strategy for natural product-based drug development against TB.

## 1. Introduction

Tuberculosis (TB), a bacterial infection caused by *Mycobacterium tuberculosis (Mtb)*, has likely regained its status as the leading cause of death from a single infectious agent worldwide, following a three-year period during which COVID-19 held that position. TB infection still remains nowadays one of the primary causes of mortality in individuals living with HIV. In 2023, the World Health Organization (WHO) reported 1.25 million deaths from tuberculosis, including 161,000 people coinfected with HIV [1]. Another major public health issue in TB treatment is the emergence of multi-drug-resistant TB strains (MDR-TB), knowing that in 2023, only about 2 out of 5 people with drug resistant MDR-TB accessed treatment [1].

Over 50% of all medications currently in clinical use are either derived from natural products (NPs) or are synthetic drugs designed based on natural pharmacophores [2,3]. Therefore, due to its huge diversity, drug discovery from NPs remains a major opportunity but also a significant challenge. The identification of novel bioactive compounds from NPs can be guided by ethnopharmacological knowledge or accomplished through systematic screening of extract libraries. However, the process of characterization and isolation of novel bioactive structures can be challenging, due to the complexity of NP extracts. The commonly used bio-guided purification process can be very time-consuming, costly, and often leads to the isolation of already described structures. To avoid these latter limitations, NP researchers have been using computational approaches, based on mass-spectrometry profiling, to identify new compounds of interest within complex mixtures. In recent studies, integrative strategies combining the use of molecular networking (MN) and the biological screening of extract libraries have been developed, prioritizing potential bioactive NPs within massive datasets. Layering multiple pieces of information on an MN can highlight structures of interest for further investigation [4,5,6,7]. The integration of structural information obtained from LC-MS/MS analysis combined with biological data from a natural extract library allows for the identification of correlations between chemical scaffolds and observed bioactivities early in the annotation process and before purification. When screening NPs for a specific bioactivity, structurally related compound families typically exhibit biological effects that are modulated by subtle structural changes in line with structure–activity relationship (SAR) principles. Accordingly, when analyzing massive MNs encompassing diverse chemical structures, bioactive compounds responsible for the activity of extracts can be uncovered by mapping bioactivity results and taxonomic information onto these networks. Recent studies, such as Olivon et al.’s work on a collection of *Euphorbiaceae* species, demonstrated the efficiency of this approach for the discovery of bioactive compounds in large datasets [5].

The present work aims to apply the multi-informational MN approach to a set of seven plant species from French Guiana in the search of novel antituberculosis agents. These seven plants, namely *Curatella americana*, *Davilla nitida*, *Dipteryx punctata*, *Indigofera suffruticosa*, *Quassia amara*, *Tetradenia riparia*, and *Zingiber zerumbet*, were selected based on their documented traditional medicinal uses in French Guiana, as well as prior studies highlighting the antibacterial activity of their extracts against *Mtb* [8,9,10].

*Curatella americana* L. (Dilleneaceae) bark is commonly used in Amazonian folk medicine for the treatment of inflammation and ulcers [11]. Decoctions of the bark are also used to treat jaundice and used as sedatives [12].

*Davilla nitida* (Vahl) Kubitzki (Dilleniaceae) is a common species in South America, and the leaves are used in infusion and decoction for a variety of applications; they are used to treat gastric pain, diarrhea, inflammation, and ulcers [13].

*Dipteryx punctata* (S.F.Blake) Amshoff (Fabaceae) is an ever-growing tree, and the bark is used by the Wayapi people as an external wash to reduce fever. It is also used by Palikur communities in a decoction mixture, combined with the crushed whole plant of *Tonina fluviatilifor*, as an infant fortifier [12].

*Indigofera suffruticosa* Mill. (Fabaceae) finds some domestic medicinal uses all across the world but especially in South America. Leaves are intensely used for inflammation [14,15].

*Quassia amara* L. (Simaroubaceae), commonly named “Couachi” or “Quinquina de Cayenne”, is an emblematic shrub in French Guiana. This species is known for its strong bitterness, which is present in all parts of the plant. Creoles use it for its fever-reducing, cholagogue, and vermifuge properties. It is also used, particularly by the Palikur people, to treat malaria and dengue fever. It is called Quinquina de Cayenne because of its bitterness and its traditional use against malaria [12].

*Tetradenia riparia* (Hochst.) Codd (Lamiaceae) is an herbaceous and aromatic shrub, commonly known as “false myrrh”. It was introduced in South America and is used as an ornamental exotic plant, especially in Brazil. While there are little to no reports of traditional uses of this species in French Guiana, it is a well-recognized medicinal plant in Rwanda and is found in nearly every rural home. The dried leaves of this plant are used as an infusion or decoction remedy against numerous diseases. It is used to treat tuberculosis symptoms, such as respiratory issues, fever, cough, and headaches [16,17].

Lastly, *Zingiber zerumbet* (L.) Roscoe (Zingiberaceae) has long been used in traditional remedies across diverse regions, notably in Southeast Asia, India, and the Pacific Islands. Known for their medicinal versatility, the rhizomes are prepared in different ways, such as decoctions, poultices, or powders, to help relieve conditions like inflammation, fever, pain, indigestion, skin conditions, and parasitic infections. Additionally, in Hawaii, the plant’s pine cones are traditionally used as shampoo, and the plant is also valued ornamentally [18,19].

Despite their frequent use in traditional medicine, little is known about the bioactive metabolites of these species, with the exception of *T. riparia*, where the diterpene 6,7-dehydroroyleanone has been described as a contributor to antitubercular activity [10]. As a result, a targeted approach based on known active constituents could not be applied, and an untargeted annotation strategy was required. In this context, MN offers a useful framework to organize and annotate complex metabolomic datasets while integrating bioactivity data to highlight candidate pharmacophores.

A key challenge in metabolite annotation is the reliance of most computational tools on spectral similarity alone, which can lead to ambiguous identifications. To strengthen annotation confidence, the recently developed taxonomically informed scoring system TIMA-R, introduced by Rutz et al. [20], was integrated in the workflow. This tool re-ranks candidate structures by combining spectral similarity with the taxonomic distance between the studied organism and known producers of candidate compounds, thereby reducing false positives and improving dereplication.

In this study, MN combined with antitubercular bioassays was applied to systematically explore the chemistry of the seven selected species (Figure 1). Correlating extract composition with bioactivity allowed the identification of metabolite families contributing to antitubercular effects, with flavonoids emerging as major candidates. To further assess the predictive value of the workflow, nine commercially available metabolites were tested, of which two flavonoids and a chlorophyll derivative displayed promising activities against *Mtb* H37Ra in the micromolar range.

## 2. Results

### 2.1. Extraction and Fractionation Process

In the context of a bioactivity-guided phytochemistry study, a standardized extraction and fractionation workflow was applied to ensure comparability across all samples. Seven different plant species were studied, with both leaves and rhizomes collected for *Zingiber zerumbet*, leading to a total of eight plant parts analyzed. Two types of crude extracts were prepared for each sample: hydro-ethanolic extracts obtained by ultrasound-assisted extraction (UAE) and aqueous extracts obtained by decoction. The crude extracts were subsequently subjected to successive liquid–liquid partitioning in order to simplify the matrices and enrich secondary metabolites.

Ultrasound-assisted extraction is a green method that avoids the use of a large quantity of solvent, offers better extraction time, and ensures good extraction yields [21]. The extraction process to obtain this library of extracts was optimized using protocols developed in a previous study [22].

For the hydro-ethanolic extracts, partitioning was performed with four solvents of increasing polarity (n-hexane, dichloromethane, ethyl acetate, and n-butanol), while for the decoction extracts, only ethyl acetate and n-butanol were used, since decoction is not suitable for extracting highly non-polar compounds. Overall, this procedure yielded two crude extracts per plant sample (UAE and decoction) and a total of seven fractions per species.

Decoction was specifically included to reflect traditional practices. In French Guiana, decoctions are widely reported for the treatment of respiratory conditions, including symptoms associated with tuberculosis. Since plant selection was partly based on such ethnopharmacological knowledge, it was important to evaluate the antitubercular potential of these traditional preparations alongside UAE extracts.

Extraction yields for both UAE hydro-ethanolic and decoction crude extracts are summarized in Appendix A.

### 2.2. Heatmaps and General Chemical Composition

The use of an untargeted LC-MS/MS analysis combined with the evaluation of the bioactivity level of each fraction or extract allowed for the construction of a feature-based MN focused on the potential bioactive features. To predict a metabolite’s bioactivity more accurately, it is important to test several samples of known concentrations that display varying related molecular profiles and bioactivity levels. These samples might include fractions obtained from a single extract or extracts from various sources [4].

The workflow developed for a bioactivity-guided MN approach can be seen in Figure 1.

For 3 (Figure 1, point 1) [23]. The use of MZMine allowed the detection and relative quantification of features (ions) across the extracts and fractions. The obtained feature list was exported both for MN and automatic annotation using the SIRIUS tool (Figure 1**,** point 2) [24]. The SIRIUS MS/MS annotation tools were used to explore the chemical diversity of the dataset. Heatmaps were generated to visualize these predictions and can be found in Figure 2. They represent the relative abundance of metabolites according to metabolic pathways, calculated from the square root of the sum of their peak areas across species, in both negative (Figure 2A) and positive (Figure 2B) ionization modes. Darker shades of red to black indicate higher abundance, while lighter yellows represent lower or no abundance. A python (3.10) script was used for the construction of the heatmaps and is available on Github [25,26].

The shikimates and phenylpropanoids pathway was dominant in all species, except for *Q. amara.* This pathway of metabolites was especially abundant in *D. punctata*, *T. riparia,* and *Z. zerumbet*. Shikimates and phenylpropanoids are mostly detected in negative ionization mode, due to the presence of hydroxyl substitutions. However, due to their high abundance in the samples, they can still be detected with a relatively high incidence in positive ionization mode. *D. nitida*, *Q. amara*, *T. riparia,* and *Z. zerumbet* showed high detection of metabolites belonging to the terpenoids pathway. Alkaloids were shown to be exclusive to *Indigofera suffruticosa* in this dataset, and were mostly detected in positive ionization mode. Lastly, *D. nitida*, *T. riparia,* and *Z. zerumbet* appeared to be rich in fatty acids. *T. riparia* also contained a lot of polyketides.

The chemical composition of the extracts’ library was investigated further by generating heatmaps based on the chemical superclasses (see Appendix A). As it could be expected based on literature, the species rich in metabolites from the shikimates and phenylpropanoids pathway appeared to be especially rich in flavonoids. *D. punctata* showed the highest detection of flavonoids and isoflavonoids across the whole dataset. *T. riparia* and *Z. zerumbet* appeared to be rich in phenylpropanoids C6–C3, while *I. suffruticosa* showed a mild detection of phenylpropanoids C6–C1, C6–C3, and flavonoids. Among the species containing metabolites from the terpenoids pathway, *T. riparia* and *Z. zerumbet* possessed abundant mono- and sesqui-terpenoids, while *C. americana* was rich in tri-terpenoids only, and *Q. amara* was rich in di- and tri-terpenoids. On the positive ionization heatmap generated according to metabolite superclasses, *I. suffruticosa* displays a high intensity of tryptophan alkaloids, followed by anthranilic acid alkaloids.

Overall, the heatmaps guided the annotation of the molecular network by highlighting relevant pathways and superclasses of compounds.

### 2.3. Bioassays Results Against Mtb H37Ra

The inhibitory activities of 72 extracts obtained from these seven medicinal plant species from French Guiana were evaluated in vitro against the growth of the attenuated *Mtb* H37Ra strain, used as a surrogate for the virulent *Mtb* H37Rv [27]. The samples exhibiting the strongest antitubercular activity can be found in Table 1, while the complete dataset, with the corresponding minimal inhibitory concentrations (MIC) leading to 50% or 90% *Mtb* growth inhibition (i.e., MIC_50_/MIC_90_), is provided in Appendix A.

Non-polar fractions (hexane and dichloromethane) exhibited the best results across all samples. More specifically, *I. suffruticosa* leaves’ hexane, dichloromethane, and ethyl acetate non-polar fractions exhibited promising antimycobacterial activity against *Mtb* H37Ra, with MIC_50_ of 244, 147, and 166 µg/mL, respectively. Their respective MIC_90_ were, however, above the tested range (>500 µg/mL). *T. riparia* leaves’ hexane and dichloromethane fractions were tested to be active with MIC_50_/MIC_90_ values of 128/>500 µg/mL and 178/425 µg/mL, respectively.

Lastly, *Z. zerumbet* leaves and rhizome extracts and fractions exhibited the best anti-TB activities. First, *Z. zerumbet* leaves’ crude extract and hexane fractions showed MIC_50_/MIC_90_ values of 32/>500 µg/mL and 211/261 µg/mL, respectively. Regarding *Z. zerumbet* rhizome, hexane and dichloromethane fractions displayed the most promising antibacterial activity, with quite good MIC_50_/MIC_90_ values of 130/253 µg/mL and 74/176 µg/mL, respectively.

The active fractions showed a relatively complex metabolite composition, as illustrated by the base peak chromatograms (BPCs) in Appendix A.

### 2.4. Bioactive Molecular Networking

The third and final step was achieved by analyzing MS/MS data on the GNPS web-platform, integrating bioassay results via metadata, and visualizing the corresponding MNs in Cytoscape v 3.8.1 (Figure 1, point 3) [28,29].

The generated MNs, both in positive and negative ionization modes, are accessible in a Zenodo repository (Zenodo repository (http://dx.doi.org/10.5281/zenodo.16086397, accessed on 25 July 2025)).

The chemical features, represented as nodes, are grouped based on their MS2 spectra similarity. Three levels of information were represented on each node, using color-coded pie charts:‑In the center, the relative abundance of the feature across the species is shown. For example, if a metabolite is detected mainly in *T. riparia*, the central color will correspond largely to that species.‑On the first ring (middle ring), the relative abundance of the feature in the extracts and/or fractions (*n*-hexane, dichloromethane, ethyl acetate, *n*-butanol and aqueous) is shown. It allows the visualization of the polarity of the metabolite.‑On the second and outer ring, the bioactivity results against *Mtb* H37Ra (relative abundance of the feature in the fractions, represented according to their activity results) are shown. The activity of each fraction is color-coded according to its MIC result, making it possible to highlight which features are enriched in active fractions.

The results of biological screening of extracts, reported as MIC_50_, were first classified according to five levels of activity (Figure 3): >500 µg/mL represents an inactive fraction, 250–500 µg/mL weakly active, 150–250 µg/mL mildly active, 75–150 µg/mL highly active sample, <75 µg/mL very highly active sample. The activity results were mapped on the external ring, to all nodes in the MN. Clusters representing features present only in highly and very highly active fractions were determined to be of main interest and annotated first.

Once the mapping of the MN was achieved and all available information were represented on the MN, the objective was to annotate detected molecules using both spectral library annotations and the network topology, in particular those molecules with significant predicted antitubercular activity.

A total of 150 metabolites were annotated across the whole dataset. Annotated compounds were classified following the levels of confidence proposed by Schymanski et al.; level 1 (L1): structure confirmed by the reference standard with MS, MS/MS spectra, and retention time matching; level 2a (L2a): probable structure using library spectrum match or literature match; level 2b (L2b): diagnostic of structure using MS/MS fragments or ionization behavior, with no literature confirmation; and level 3 (L3): tentative candidates with uncertainties (for example, positional isomers) [30]. Various reference standards, including commercial molecules tested for their antitubercular activity, were analyzed by LC-MS/MS to confirm their identification (level L1, according to the Schymanski classification). All described compounds can be found in Appendix A. Spectral data acquired in both positive and negative ionization modes were used for the annotation process. All compounds described in Appendix A have been proposed and/or verified through manual annotation.

Amongst clusters displaying interesting bioactivity levels, some were common to multiple species. Their co-occurrence in the samples showing the best antitubercular activities could make them bioactive candidates. However, the taxonomic mapping showed that the studied species have quite specific chemical composition, and there was very little overlap in the ions detected (as it might be expected for plant species of different genera). The step of fractionation was then really useful, as it allowed the discrimination of the potentially active metabolites specific to each species. If a feature is unique to a very highly active sample, it could be responsible for the bioactivity, as it is different from ubiquitous compounds. The use of the GNPS web-platform [29] helped dereplication, and the use of the spectral library helped the identification of known secondary metabolites to move further and focus the identification on novel structures. Overall, 3% in negative ionization mode and 8% in positive ionization mode of the MNs could be annotated using experimental DB spectra from GNPS. Therefore, computational tools were important to complete the annotation as exhaustively as possible. SIRIUS was used for in silico annotation, and the TIMA-R tool was implemented to improve its annotation results quality.

For our dataset, TIMA-R was implemented to re-rank SIRIUS annotations [20]. As an example, boronolide (feature ID 5676 on the positive MN), described in *T. riparia*, was initially ranked as the 10th compound proposition by SIRIUS. This feature could therefore not be annotated based on SIRIUS alone, as the first propositions appeared to be irrelevant. It was reranked first after TIMA-R and could therefore be annotated with high confidence.

This tool helped prioritize biologically relevant candidates, reduce false positives in in silico annotation, and improve dereplication in a large dataset.

Annotation of the potentially active clusters led to mainly polyphenolic structures, especially flavonoids. The clusters of flavonoids displayed diverse bioactivity levels, as some were mainly colored in red, so they were inactive, and others in green, as potentially active to highly active. This allowed for the discrimination of active compounds amongst structurally similar scaffolds.

A detailed examination of the whole MN allowed the annotation of mainly two families: terpenoids, especially di- and tri-terpenoids, and polyphenolic compounds. Among the abundant terpenoids in *Q. amara*, a significant cluster of quassinoids, part of the tri-terpenoids family, could also be annotated, as they were first discovered in this particular species [31]. On the positive MN, quassin and analogs such as neoquassin were annotated on the main cluster specific to the species. A big cluster of tri-terpenoids found mainly in *C. americana*, *D. nitida,* and *T. riparia* was also annotated and contained metabolites such as oleanolic acid, ursolic acid, and tri-terpenoids. However, these terpenoids displayed little to no bioactivity, based on the screening results of the corresponding extracts. They were therefore not looked more upon, as these clusters were not of main interest following our bioactive MN approach. Amongst polyphenolics compounds, flavonoids were predominant across the whole dataset, with multiple subclasses of flavonoids represented. For example, *C. americana* was shown to be rich in flavanols, precisely in catechin and derivatives, while *D. punctata* contained various isoflavones, such as pseudobaptigenin and derivatives. Flavones, especially methoxyflavones, were the major compounds in the most active fractions. *Z. zerumbet* roots and leaves were shown to be rich in afzelin (a glycosyloxyflavone) and its acetylated derivatives, such as afzelin acetate and afzelin diacetate [32]. *T. riparia* was shown to be rich in hispidulin (a monomethoxyflavone), cirsimaritin, and cirsiliol (dimethoxyflavones). An important cluster specific to *T. riparia* was also annotated as boronolide and derivatives on the positive MN. Based on the fragmentation patterns and multiple adducts, with [M + Na]^+^ and [M + NH_4_]^+^ as the most abundant, boronolide and deacetylboronolide were identified as the main compounds in the active fractions of *T. riparia*.

### 2.5. Validation of Selected Identified NPs

To confirm whether the MN was informative and could predict with a relatively good accuracy the potential bioactivity of a metabolite, nine commercially available compounds were purchased and tested against *Mtb* H37Ra following the same protocol as for the extract library. These compounds were selected to represent the different categories of clusters: metabolites annotated on red-colored clusters, indicating potential inactivity of the compound and its analogs; and metabolites annotated on yellow- to green-colored clusters, indicating potentially active to very active clusters. The clusters corresponding to metabolites evaluated for their antitubercular activity can be seen in Figure 4A,B for the negative and positive ionization MN, respectively, and their corresponding determined MIC values can be found in Table 2.

On the negative ionization mode MN (Figure 4A), the bioactivity prediction appeared to be accurate; casticin was on an inactive cluster, while hispidulin and cirsiliol were on active clusters. On the positive ionization mode MN (Figure 4B), it appeared that the clusters indicated as bioactive contained more false positive hits. This resulted in the testing of inactive compounds: zerumbone and caryophyllene oxide displayed very low to no activity, despite being annotated on potentially active to very active clusters. This could be explained by the presence of these compounds in fewer samples than the flavonoids, which are omnipresent in plant samples. While spectral features that are detected in only one species and in one fraction type could be of potential interest against tuberculosis, the absence of data overlapping between samples also make these clusters at a higher risk of false positives.

The boronolide and analogs clusters were also colored in yellow/light green on the positive MN, indicating metabolites potentially contributing to the antitubercular activity. In previous studies, no activity was reported for boronolide and analogs. Further purification and activity evaluation of boronolide and its analogs could confirm or invalidate these predictions. The antitubercular activity of *T. riparia* has been previously attributed to 8(14),15-sandaracopimaradiene-7α,18diol (MIC_90_ = 12 µg/mL against *Mycobacterium smegmatis*), a diterpenediol described for the first time by Van Puyvelde et al., and 6,7-dehydroroyleanone (MIC_90_ = 31.2 µg/mL against *Mtb* H37Rv), isolated from the essential oil of *T. riparia* leaves by Baldin et al. [10,33]. On the negative ionization mode MN, 6,7-dehydroroyleanone was annotated as a minor compound in the studied samples of *T. riparia*. However, the diterpenediol 8(14),15-sandaracopimaradiene-7α,18 diol could not be detected in the studied samples. The hypothesis is therefore that the antitubercular activities of *T. riparia* leaves’ hexane and dichloromethane fractions are due to a mixture of compounds; flavonoids such as hispidulin and cirsiliol; and terpenoids such as α-bisabolol and 6,7-dehydroroyleanone. For *I. suffruticosa*, heatmaps (see Appendix A) revealed a high incidence of tryptophan alkaloids. On the positive MN, SIRIUS predictions, re-ranked by the TIMA-R tool, identified a cluster specific to *I. suffruticosa* containing carboline alkaloids. No metabolite could be confidently identified, based on experimental databases. Purification of the active dichloromethane fraction of *I. suffruticosa* leaves could be performed in the future to help refine the identification of these metabolites.

### 2.6. Cytotoxicity Assay Results

Following the determination of the antitubercular activity with the nine selected NPs, five compounds were selected for further evaluation of their toxic effects. Cytotoxicity assays were carried out on two non-tumorigenic human cell lines: BEAS-2B (bronchial epithelial cells) and IMR90 (embryonic lung fibroblasts). The CC_50_, defined as the concentration leading to a 50% decrease in cell viability, was then determined for each compound, and the therapeutic index (TI), defined as the ratio of CC_50_ to the MIC_90_ against *Mtb* H37Ra (see Table 3), was calculated when applicable.

Although α-bisabolol and zerumbone showed no activity against *Mtb* H37Ra up to a 200 µM threshold, these two molecules were also tested for cytotoxicity. No toxicity was detected for α-bisabolol up to a 250 µM concentration in BEAS-2B, IMR90, and HepG2 cells, thereby indicating a favorable safety profile. Zerumbone, despite being nontoxic on HepG2 up to 250 µM, displayed significant cytotoxicity in both BEAS-2B and IMR90, with CC_50_ values of 149 µM and 79.8 µM, respectively. The associated TIs (>1.25, <0.74 and <0.4, respectively) raised concerns regarding its therapeutic potential. Cirsiliol was found to be nontoxic to BEAS-2B and HepG2 cells (CC_50_ > 250 µM) but moderately toxic to IMR90 (CC_50_ = 107.6 µM), yielding a low TI < 0.54. Hispidulin, exhibiting promising activity against *Mtb* H37Ra, showed low to no toxicity, with CC_50_ values of 145.9 µM (IMR90) and >250 µM (BEAS-2B and HepG2), corresponding to a TI > 0.9 for IMR90 and >1.5 for both BEAS-2B and HepG2. Finally, pheophorbide A was the most cytotoxic compound, with CC_50_ values of around 4.9 µM and 11.5 µM when exposed to BEAS-2B and IMR90, respectively, resulting in a poor therapeutic index (TI = 0.29–0.68). Nevertheless, this molecule displayed no cytotoxicity to HepG2 cells.

In addition, all compounds appeared to be cytotoxic against two cancerous macrophage cell lines: Raw264.7 murine macrophage cells and THP-1 human-derived macrophages [34,35,36,37,38] (see Appendix A).

## 3. Discussion

This study presents a comprehensive bioactivity-guided phytochemical investigation of seven medicinal plants from French Guiana, integrating untargeted LC-MS/MS-based metabolomics, molecular networking, and antitubercular bioassays to identify bioactive metabolites against *Mtb* H37Ra.

The UAE and decoction extraction methods allowed for a broad coverage of metabolites, from polar polyphenolic compounds to non-polar terpenoids. The extracts and fractions obtained by decoction showed no or very weak antibacterial activity. UAE, recognized for its efficiency in extracting secondary metabolites, combined with successive liquid–liquid partitioning, allowed for a detailed exploration of the chemistry of each species. This extraction process was very suitable for the detection of non-polar compounds, which appeared to be mainly responsible for the antitubercular activity. Indeed, our results not only confirm the antimycobacterial potential of some of these species but also provide novel insights into their chemical composition and active constituents, aligning with, or in some cases challenging, previous reports. The developed method demonstrated that the bioactive MN approach can discriminate bioactive flavonoids (such as hispidulin, MIC_50_ = 54.8 µM) from inactive analogs (such as casticin, MIC_50_ > 200 µM), even within structurally similar clusters. This underscores MN’s utility in prioritizing ubiquitous compounds for targeted testing, bypassing laborious isolation steps.

This approach also highlights the advantages of MN when compared with the more conventional bioactivity-guided fractionation strategy. Classical fractionation remains indispensable for the purification and full structural elucidation of novel metabolites, but it is known to be labor-intensive, time-consuming, and often leads to the repeated isolation of compounds already described. By contrast, MN makes it possible to prioritize bioactive candidates at an earlier stage by combining chemical and biological information directly within complex datasets. This approach limits redundancy and helps distinguish active from inactive analogs, even within families of widely distributed metabolites, such as flavonoids. A drawback, however, is that MN relies heavily on in silico predictions, which must ultimately be confirmed by isolation and spectroscopic characterization. For these reasons, the two strategies are best viewed as complementary: MN provides a rapid means of highlighting promising candidates, while bioactivity-guided fractionation remains essential for validation and structural confirmation, especially for novel structures.

Results showed on the heatmaps agree with literature data, especially for *Q. amara,* in which triterpenoid lactones of the quassinoids class have been extensively described [31]. Conversely, no terpenes have been previously described in *D. nitida* via LC-MS studies. In *T. riparia*, few studies report the chemical composition of non-volatile liquid extracts, with only two main diterpenes isolated in the leaves, ibozol and a diterpenediol, described for the first time in the species [33,39]. In the *Indigofera* genus, multiple alkaloids of the anthranilic acid superclass have been described, such as indigo, the most commonly used pigment in the world [40]. However, only one tryptophan alkaloid, indican, has been previously described, despite a high detection of this superclass in the generated heatmaps [41].

The MIC determination highlighted three species with notable antibacterial activity: *Z. zerumbet*, *T. riparia,* and *I. suffruticosa*.

Both *Z. zerumbet’s* leaves and rhizome showed interesting activity. These results are in good agreement with the work of Carrion et al. that was performed using a resazurin microtiter assay plate [42]. In this study, *Z. zerumbet* leaves’ dichloromethane extracts exhibited good activity against three strains of *Mtb*, two of them being resistant strains, with MIC_90_s ranging from 12.5 µg/mL to 200 µg/mL [9]. However, our UAE, followed by liquid–liquid partitioning, revealed the hexane fraction as the most bioactive, suggesting that active compounds in Carrion’s et al. dichloromethane extracts may partition differently under our protocol. *Z. zerumbet* rhizome hexane and dichloromethane fractions displayed the best antitubercular activities, with MIC_90_s in good agreement with the work of Phongpaichit et al. who reported an MIC_90_ of 125 µg/mL for the chloroform extract against *Mtb* H37Ra [43]. In this work, curcuminoids, namely curcumin and methoxycurcumin, were proposed as potentially responsible for the antitubercular activity. However, they were not highlighted as potential contributors to the activity on our MNs. The negative ionization MN highlighted flavonoids, such as afzelin and derivatives.

*T. riparia* was the second species displaying interesting antibacterial activities. Among multiple studies describing antitubercular activity for *T. riparia*, Van Puyvelde et al. and Baldin et al.’s works reported interesting antituberculosis activity from the leaves’ chloroform extract and the essential oil, which reinforces our own results [10,33]. More precisely, in the study from Van Puyvelde et al., a diterpene, 6,7-dehydroroyleanone, was described as responsible for the antitubercular activity of the leaves’ essential oil (EO). This compound was annotated in the most active fraction of *T. riparia* in our dataset. However, the MN proposed two flavonoids (hispidulin and cirsiliol) as additional contributors, expanding the known bioactive metabolites of the species.

Lastly, the *I. suffruticosa* dichloromethane fraction exhibited a notable activity, confirming results found in previous studies on methanolic extracts of *I. suffruticosa* leaves [8,44]. However, in the study published in 2010 by Carli et al., the dichloromethane fraction had not shown any antitubercular activity using the microplate Alamar Blue assay (MABA) [44]. The difference in both results could be explained by the use of different protocols.

Overall, across the whole dataset, features predicted as bioactive metabolites appeared to be mainly flavonoids. Various flavonoids have been extensively studied for their antimycobacterial properties and have shown interesting activities [45,46,47]. Quercetin, for example, showed interesting antitubercular properties, with an MIC_90_ of 6.25 µg/mL against H37Rv [47,48]. The obtained MN seemed to predict some specific types of flavonoids as having higher bioactivity against *Mtb*. In literature, studies have found that less common flavonoid scaffolds, such as afrormosin or formononetin, displayed interesting activities against *Mtb* H37Ra, with MIC_90_, respectively, of 50 and 25 µg/mL [49]. Some flavonoids that have been previously reported in literature with notable bioactivities could be annotated on the MN on bioactive clusters, as can be seen in Figure 5. Confronting these bioactivity results with our MN confirmed its predictive capacity; as an example, naringenin appeared to be most active on the cluster, with a measured MIC_90_ < 2.8 µg/mL against *Mtb* H37Rv [46].

The focus was then to evaluate if the MN discriminated active from non-active metabolites amongst very ubiquitous compounds such as flavonoids. These structures, very well described and abundant in plants, can easily be annotated through public databases. If the bioactive MN approach differentiates active from non-active compounds amongst ubiquitous families, they can then easily be purchased and tested against the strain, without the need for a purification step. This could allow for the discovery of unexpected activities amongst already described structures. Amongst tested commercial compounds, cirsiliol and hispidulin were predicted as mildly active and indeed showed medium activity, with MIC_50_/MIC_90_ of 80.3/> 200 µM and 54.8/162 µM. In the same chemical superclass, casticin was predicted as inactive and displayed no activity (MIC > 200 µM). For flavonoid structures, detected mainly in negative ionization mode, the MN could accurately predict the antitubercular activity of a compound, even between structurally similar metabolites. However, for terpenoid structures, detected in positive ionization mode, the MN predictions showed lower accuracy. For example, α-bisabolol showed no antitubercular activity (MIC_50_ > 200 µM), despite being predicted as active on the MN. Moreover, caryophyllene oxide, loliolide, lyoniresinol, and zerumbone were also found inactive, despite being predicted as mildly active on the MN.

For the studied plant species, results could be found in literature describing interesting activities, mainly on crude extracts and against H37Rv strains of *Mtb.*

Not all activities shown in literature could be confirmed; for most studied species, very low to no activity was found, despite good MIC described in literature. For example, in a study by Pavan et al., *C. americana* bark’s chloroform extract was evaluated as active against *Mtb* H37Rv, with an MIC_90_ of 62.5 µg/mL. In this same study, *D. nitida* leaves’ chloroform extract showed activity, with an MIC_90_ of 125 µg/mL [8]. The difference in results could be explained by multiple factors: difference in provenance, plant material collection period, use of a different strain, and/or different extraction protocols.

While the negative ionization mode MN reliably predicted flavonoid activity, false positives in positive ionization mode (such as α-bisabolol, zerumbone) highlight a limitation: rare features may lack sufficient data for accurate bioactivity mapping. This suggests that MN predictions require validation, particularly for species–specific metabolites. To reduce false positives and resolve unique bioactive metabolites, future work should employ high-resolution fractionation (HPLC sub-fractionation) of active extracts. This would help refine MN clusters to move past ubiquitous compounds and isolate minor metabolites, such as the uncharacterized carboline alkaloids in *I. suffruticosa*.

Lastly, we investigated the cytotoxic effect of the most promising compounds to determine their therapeutic relevance. Cytotoxicity assays were conducted on two non-carcinogenic cell lines (BEAS-2B and IMR90) and three cancerous cell lines (Raw264.7, THP-1 macrophages, and HepG2). Overall, α-bisabolol and hispidulin exhibited favorable safety profiles, with CC_50_ values above 250 µM to BEAS-2B and HepG2 and relatively low cytotoxicity to IMR90. Cirsiliol and zerumbone exhibited moderate to significant cytotoxicity, depending on the cell line, leading to a low therapeutic index. Unfortunately, pheophorbide A, despite being the most active compound against *Mtb* H37Ra growth and showing no toxicity against HepG2 up to 84.3 µM, showed strong cytotoxicity across all tested cell lines, especially immune cells, making it unsuitable for further development without structural modification or targeted delivery strategies. In literature, α-bisabolol, cirsiliol, hispidulin, pheophorbide A, and zerumbone have been reported to exhibit anticancer activities. For example, α-bisabolol has demonstrated pro-apoptotic effects in human acute leukemia cells [38] and A549 NSCLC cells (non-small lung cancer cells) due to G2/M cell cycle arrest and mitochondrial apoptosis [35]. Cirsiliol inhibits HCT116 and SW480 cell (colon cancer cells) mitophagy by targeting STAT3 [34]. Hispidulin induces apoptosis of human hepatoblastoma cancer cells (HepG2) in a dose-dependent manner through mitochondrial dysfunction and inhibition of the PI3k/Akt signaling pathway [36]. Pheophorbide A has an antiproliferative activity against multiple cell lines like MES-SA-human uterus from uterine carcinoma, Hep3B-human liver from hepatocellular carcinoma, or U87MG human brain from likely glioblastoma [37]. Here, we demonstrate that hispidulin, also identified as an inhibitor of the *Mtb* proteasome required for bacterial virulence [50], offers an attractive balance of in vitro efficacy against *Mtb* H37Ra and safety for non-malignant cell lines. In contrast, the therapeutic window of other potent compounds, such as pheophorbide A, is narrow. In silico ADME study of hispidulin shows high oral absorption, good solubility, and full compliance with all drug-likeness rules, making it a strong candidate for drug development. It is free from PAINS and Brenk alerts, with moderate lipophilicity (LogP ≈ 2.1) and a favorable bioavailability score (0.55), while limited BBB permeability lowers risk of CNS side effects (useful, since TB therapy is long). The main limitation of this compound is the potential CYP450 inhibition (1A2, 2D6, 3A4), which may cause drug–drug interactions but remains manageable through chemical optimization.

Although MIC and cytotoxicity assays were performed for a subset of annotated compounds, several candidates (such as boronolide) were not tested directly due to limited availability or intermediate annotation confidence. Compound selection was therefore dictated by accessibility and predictive strength within prioritized clusters. Nonetheless, the observed variation in activity across structurally related metabolites suggests that structure–activity relationships (SAR) are at play—for example, differences in MIC between flavonoids such as hispidulin and casticin. Previous studies on flavonoids have demonstrated the importance of functional group positioning for their antibacterial activity: hydroxyl and prenyl substitutions, lipophilicity, and ring substitution pattern [51,52].

Similarly, QSAR modeling has been successfully applied to chalcone- and flavonoid-based antimycobacterial compounds using genetic function approximation (GFA) to identify the molecular descriptors most related to the antitubercular activity [53]. Although QSAR or SAR work was not performed in this study, the MNs generated provide an interesting foundation for such computational modeling. Implementing QSAR studies in future works could allow for a more precise prediction of potential new bioactive compounds.

## 4. Materials and Methods

### 4.1. Plant Material

Plant material was collected in Cayenne, French Guiana, by BioStratège S.A.S. Taxonomic identification was performed in Cayenne by Pierre Silland for all species, except *T. riparia*, which was identified by Elodie Desmarest (farmer at Declic Jardin, French Guiana). All collected species are wild, except *Q. amara*, which is cultivated. Voucher specimens were deposited at the Cayenne University herbarium. Collection information and registered barcodes can be found in Table 4. The species collection location has been shared through CardObs [54].

All samples were dried in a ventilated space, then crushed into powder using a knife mill (cutter mill) and packed in sealed containers for transport.

### 4.2. Extracts Library

#### 4.2.1. Crude Extracts Preparation

An ultrasound-assisted extraction process was used for the production of crude extracts, with 50 g of the dry plant powder weighed and added directly into the extractor. A volume of 500 mL 70/30 (*v*/*v*) of ethanol and water mixture was added to the extractor, and the resulting mixture was sonicated with ultrasounds for 30 min at 25 °C (PEX05 25 kHz, Reus France). The mixture was agitated during the whole process, and the extractor was filled with nitrogen gas to limit oxidation during the extraction process. Three analytical replicates were prepared for each species. The resulting mixtures were filtered under vacuum. The obtained extracts were dried under vacuum using a rotary evaporator, and yield extractions were determined. A second crude extract was prepared for each plant sample using a decoction method: 10 g of the dry plant powder was weighted and added to a beaker with 150 mL of distilled water. The mixture was heated on a heating plate for 15 min at 100 °C. Magnetic agitation was performed during the whole process. The resulting mixtures were filtered under vacuum. The obtained extracts were freeze-dried (Cryotec, Lunel Viel, France), and yield extractions were determined.

#### 4.2.2. Fractionation of Crude Extracts

Fractionation of crude extracts was performed by successive liquid–liquid partitioning. Hydro-ethanolic extracts were fractioned using four solvents of increasing polarity: *n*-hexane, dichloromethane ethyl acetate, and *n*-butanol. The residual aqueous fractions were kept. Decoction extracts were also fractioned using two solvents of increasing polarity: ethyl acetate and *n*-butanol. As a result, five fractions were obtained per hydro-ethanolic extract and two per decoction extract. The obtained organic fractions were then dried under vacuum using a rotary evaporator, while aqueous fractions were freeze-dried. Fractionation yields were determined.

#### 4.2.3. Extracts and Fractions Sampling

In total, 72 different samples (extracts and fractions) were obtained, each in analytical triplicates. Dried samples were solubilized in appropriate solvent (methanol or acetone, according to the polarity of the fraction/extract) at a known concentration of 20 mg/mL, then sampled in 96-well plates. The plates were dried under nitrogen and stored at −20 °C for conservation and further analysis. Considering the large number of samples generated, a codification was established as followed:XX-P-E-0-FR

With:XX = Plant species (first letter of the genus and first letter of the species)P = plant part (B = bark; L = leaves; W = wood; R = rhizome)E = extract type (E = UAE = ultrasound-assisted extraction; D = decoction)0 = replicate number (ranging from 1 to 3);FR = fraction solvent (if the sample is a fraction; HE = hexane; DM = dichloromethane; EA = ethyl acetate; BU = butanol; AQ = aqueous).

*Example*: TR-L-E-1-DM is the dichloromethane fraction obtained from the fractioning of the crude hydro-ethanolic extract, replicate 1, of *T. riparia* leaves.

### 4.3. HPLC-HRMS/MS Analysis

Extracts and fractions were filtered using 0.22 µm PTFE filters (Restek, Lisses, France) into glass vials at a concentration of 5 mg/mL for crude extracts and 2 mg/mL for fractions.

Commercial standard compounds were also filtered using 0.22 µm PTFE filters (Restek, Lisses, France) into glass vials at a concentration of 0.1 mg/mL.

The LC-MS/MS analysis was performed on a Thermo Scientific Dionex 3000 Ultra-High-Performance Liquid Chromatography system (UHPLC) coupled to a Bruker Impact II Q-TOF high-resolution mass spectrometer equipped with an electrospray ionization source (ESI). The chromatographic separation was carried on an Agilent Zorbax Eclipse Plus C18 column (2.1 × 100 mm, 1.8 µm) at 43 °C. Ultrapure water (A) (LC-MS grade, Carlo Erba, Italy) and acetonitrile (B) (LC-MS grade, Carlo Erba, Italy), both acidified with 0.1% formic acid (LC-MS grade, Carlo Erba, Italy), were used as mobile phases. The injection volume was 1 µL for all samples, and the flow rate of the mobile phase was 0.8 mL/min. The following gradient was applied: isocratic hold at 5% B for 2 min, 5–30% B over 2–17 min, 30–50% B over 17–27 min, 70–100% B over 27–31 min, then isocratic hold at 100% B for 1 min (31–32 min), followed by a decrease to 5% B in 0.1 min (32–32.1 min), and held at 5% B over 32.1–34 min for the column equilibration for the next experiment.

Mass spectrometry analyses were conducted in both positive and negative electrospray ionization modes, covering a *m*/*z* range of 50 to 1200. For each mode, the Q-TOF instrument was operated with the following settings: end plate offset at 500 V, nebulizer nitrogen pressure maintained at 3.5 bar, drying gas (N_2_) flow rate at 12 L/min, and a drying temperature of 200 °C. Data were acquired at a rate of 4 Hz. The capillary voltage was set to 3500 V in positive mode and 3000 V in negative mode. A data-dependent acquisition (DDA) method was employed; MS/MS fragmentation spectra were automatically generated for the three most intense precursor ions using a stepped collision energy ranging from 20 to 40 eV. The precursor ion isolation width was set to 5 *m*/*z* (2.5 from each side of precursor *m*/*z*). A solution of sodium formate acetate was used as a calibration to obtain high mass accuracy (<5 ppm) and was automatically injected at the beginning of each run. A calibration mixture was used to correct potential retention time drifting: a mixture of parabens with an increasing carbon chain from hydroxybenzoic acid to decyl paraben were used.

The nine commercial molecules were analyzed using the same method as the plant samples to confirm the annotations. All nine compounds, casticin, cirsiliol, hispidulin, lyoniresinol, loliolide, α-bisabolol, caryophyllene oxide, zerumbone, and pheophorbide A were purchased from MedChemExpress (Monmouth Junction, NJ, USA).

### 4.4. Data Treatment

#### 4.4.1. File Conversion

Raw datasets obtained from the UHPLC-MS/MS system were calibrated using Bruker DataAnalysis (5.0 SR1 64-bit) and converted into open format.mzXML using the MS Convert tool from Proteowizard [55].

#### 4.4.2. Data Processing

Exported.mzXML data were pre-processed using MZmine software, version 3.9.0. The processing workflow included raw data file import, mass detection, chromatogram building, smoothing, chromatogram resolving, feature list deisotoping, alignment between analytical replicates, filtering, alignment across all samples, and filtering across all samples [29,56,57]. The features present in the blank solvent runs were removed from the features list. Parameters of each step used for processing can be seen in Appendix A of Appendix A.

#### 4.4.3. Feature-Based Molecular Networking

Processed data were exported (mgf and CSV files) from MZmine and uploaded onto the GNPS platform for feature-based molecular networking [29]. Precursor mass ion tolerance and MS/MS fragment ion tolerance were set to 0.02 Da. The molecular network was generated with an edge cosine score above 0.7 and minimum six matched peaks between MS2 spectra. An edge was formed between two nodes if they were in each other’s top 10 most similar nodes. The maximum size of a cluster was set to 100 nodes.

The MS2 spectra in the network were searched against GNPS public spectral libraries. The resulting network was visualized using Cytoscape. Raw data of LC-MS/MS analysis were deposited in the MassIVE Public GNPS dataset. The molecular networking job on GNPS can be found at https://gnps.ucsd.edu/ProteoSAFe/status.jsp?task=a2ed23823eed46fcb6b1081735588de9 (negative ionization mode job, accessed on 10 July 2025) and https://gnps.ucsd.edu/ProteoSAFe/status.jsp?task=8ee98816e7ff4b1aad3ede4355266b73 (positive ionization mode job, accessed on 10 July 2025). The corresponding Cytoscape file is available as Appendix A.

#### 4.4.4. SIRIUS

SIRIUS implements computational tools such as CANOPUS to categorize chemical compounds using mass spectrometry data. CANOPUS can function on its own system but can also reference other classification systems, such as NPClassifier, an ontology based on 7 major metabolic pathways, divided in 70 superclasses and 672 individual classes [58].

A total of 8466 features in positive mode and 15,691 features in negative mode were submitted to SIRIUS for automatic annotation [24]. The following parameters were set: instrument (*Q*-TOF), filter by isotope pattern, MS^2^ mass accuracy = 10 ppm, molecular formula candidates stored = 10, and minimum candidates per Ion stored = 1. The following adducts were considered: [M + H]^+^, [M + K]^+^, [M + Na]^+^ in positive mode; [M-H]^−^, [M + Cl]^−^ in negative mode. Default parameters were kept for the ILP module and elements-allowed module. The following databases were selected both for formulas determination and CSI:FingerID structure elucidation: CHEBI, COCONUT, GNPS, KNApSAcK, Maconda, NORMAN, Natural Products, and Plantcyc. Intern databases were also added, based on the metabolites described in literature for the studied species. The following fallback adducts were selected for CSI:FingerID structure elucidation: [M + H]^+^, [M-H_2_O + H]^+^, [M + NH_3_ + H]^+^, [M + H_2_O + H]^+^, [M + CH_4_O + H]^+^, [M + K]^+^, and [M + Na]^+^ in positive mode and [M-H]^−^, [M + Cl]^−^, and [M + Na-2H]^−^ in negative mode. ZODIAC and CANOPUS modules were also selected. (Default parameters were kept.) A total of 33 metabolite features in positive ionization mode and 273 metabolite features in negative ionization mode were excluded from the analysis due to a molecular mass above 850 Da.

#### 4.4.5. Taxonomic Informed Scoring

The TIMA-R tool was used to re-rank candidates proposed by SIRIUS according to taxonomic information of the studied samples [20]. The obtained exports from MZmine, SIRIUS, and TIMA-R were uploaded to the Zenodo repository (http://dx.doi.org/10.5281/zenodo.16086397, accessed on 25 July 2025).

### 4.5. Antimycobacterial Activity Against Mycobacterium Tuberculosis H37Ra

Each extract and fraction were resuspended in either sterile water or DMSO, depending on their solubility. *Mtb* H37Ra strain (ATCC 25177) was routinely grown at 37 °C under stirring (60 rpm) in Middlebrook 7H9 broth (#271310, BD Difco, Le Pont de Claix, France) supplemented with 0.2% glycerol (#EU3550, Euromedex, Souffelweyersheim, France), 0.05% Tween-80 (#P1754, Sigma-Aldrich, Saint-Quentin Fallavier, France), and 10% oleic acid, albumin, dextrose, and catalase (#211886, OADC enrichment; BD Difco) (7H9TG^OADC^).

The different extracts and fractions were screened for their antibacterial activity against *Mtb* H37Ra using the broth microdilution assay in 96-well flat-bottom Nunclon Delta Surface microplates with the lid (#167008, Thermo-Fisher Scientific, Illkirch, France) [59,60]. Briefly, log-phase bacteria were diluted to a cell density of 5 × 10^6^ CFU/mL in complete 7H9TG^OADC^ medium. Then, 100 μL of the above inoculum was added to each well containing 100 μL of complete 7H9TG^OADC^ medium, serial two-fold dilutions of the compounds or controls, to a final volume of 200 μL (final bacterial load of 5 × 10^5^ CFU per well). Growth controls containing no compound or with the DMSO vehicle (i.e., bacteria only = *B*), inhibition controls containing 50 μg/mL kanamycin (#UK0010D, Euromedex), and sterility controls (i.e., medium only = *M*) without inoculation were also included. Microplates were incubated at 37 °C in a humidity chamber to prevent evaporation for 14 days, followed by absorbance reading at 600 nm using a Tecan Spark 10M™ multimode microplate reader (Tecan Group Ltd., Männedorf, Switzerland). Following a background subtraction on all wells with a mean of *M* wells, relative growth was defined as: RG% = (test well OD_600_/mean OD_600_ of control *B* wells) × 100. Minimal inhibitory concentrations (MIC) were further determined by fitting the RG% sigmoidal dose–response curves in Kaleidagraph 4.2 software (Synergy Software, Reading, PA, USA). The lowest compound concentrations leading to 50% or 90% inhibition of bacterial growth were defined as the MIC_50_ and MIC_90_, respectively. All experiments were performed independently at least three times in duplicate. All samples were evaluated except for the residual aqueous fraction obtained from the decoction extract. This is due to low extraction yields not creating enough sample to perform the bioassays. Therefore, in total, 72 samples were tested for their activity against *Mtb* H37Ra. Nine commercial compounds were tested against *Mtb* H37Ra following the same method to assess their antitubercular potential.

### 4.6. Cytotoxic Assay on Mammalian Cells

The cytotoxicity of five selected commercial compounds was evaluated using the resazurin reduction assay, as previously described [59].

Human bronchial epithelial cells BEAS-2B (ATCC CRL-9609), human liver cell line HepG2 (ATCC HB-8065), human fetal lung fibroblasts IMR-90 (ATCC CCL-186), Raw264.7 murine macrophages cell line (ATCC TIB-71), and THP-1 human-derived macrophages (ATCC TIB-202) obtained from the American Type Culture Collection (ATCC) (Molsheim Cedex France) were used.

The THP-1 human acute monocytic leukemia cell line was cultivated at 37 °C with 5% CO_2_ in RPMI-1640 medium (ThermoFisher Scientific Invitrogen, Illkirch–Graffenstaden, France) supplemented with 10% heat-inactivated fetal bovine serum (FBS, ThermoFisher Scientific Invitrogen) (RPMI^FBS^) in 75 cm^2^ flasks. Upon reaching confluence, THP-1 monocytes were seeded into 96-well flat-bottom microplates (Greiner Bio One Cellstar from Dominique Dutscher, Brumath, France) with a lid at a density of 1.5–2 × 10^4^ cells/well in RPMI^FBS^ and differentiated into adherent macrophages in the presence of 10 ng/mL PMA (Phorbol 12-Myristate 13-Acetate-Sigma-Aldrich, P8139) for 72 h, then maintained for 24 h at 37 °C with 5% CO_2_ in RPMI^FBS^ without PMA before exposure to molecules. Murine macrophages Raw264.7, human fetal lung fibroblasts IMR-90, human liver from hepatocellular carcinoma HepG2, and human bronchial epithelial cells BEAS-2B were cultured in DMEM supplemented with 10% heat-inactivated FBS (DMEM^FBS^) in 75 cm^2^ flasks to subconfluent concentrations. Around 1.5–2 × 10^4^ cells/well were seeded in 96-well flat-bottom microplates (Greiner Cellstar) with a lid in fresh DMEM^FBS^ medium and cultured for an additional 48–72 h at 37 °C and 5% CO_2_ until cell confluence.

The medium was removed, and 100 μL of serial two-fold dilutions of the compounds in RPMI^FBS^ (for THP-1-derived macrophages) or DMEM^FBS^ (for Raw264.7 macrophages, BEAS-2B, HepG2, and IMR-90 cells) were added to each well. Following a 24 h incubation period at 37 °C and 5% CO_2_, the medium was removed, and cell viability was assessed using resazurin (from Sigma Aldrich, reference 199303) diluted in PBS^++^ at 30 µg/mL (final concentration).

One hundred microliters of this solution were added to each well, and after 1 h of incubation at 37 °C and 5% CO_2_ in the dark, fluorescence intensity was measured using a microplate reader (SynergyMx, BioTek, Colmar, France; λ_ex_/λ_em_ = 530/590 nm). Cells treated with DMSO (vehicle control) were used as reference for 100% viability, with Triton X100 being used as the positive control of toxicity. The cytotoxic concentration of each compound causing a 50% reduction in cell viability, as compared to the control, was defined as the CC_50_ and was determined by fitting the relative fluorescence unit (RFU%) sigmoidal dose–response curves in KaleidaGraph 4.2 software (Synergy Software, Reading, PA, USA). All experiments were performed independently three times in duplicate.

### 4.7. Pharmacokinetics Study In Silico

Swiss ADME was used for in silico ADME studies (http://www.swissadme.ch/, accessed on 22 September 2025).

## 5. Conclusions

This study demonstrates the effectiveness of a bioactivity-guided molecular network approach in accelerating the discovery of antitubercular compounds from natural products. By integrating high-resolution LC-MS/MS data with in vitro bioassays, the workflow enabled the identification and prioritization of bioactive metabolites within a chemically diverse library of 72 extracts and fractions derived from seven plant species native to French Guiana. Among the most promising species, *Zingiber zerumbet*, *Tetradenia riparia*, and *Indigofera suffruticosa* exhibited notable activity, primarily in their non-polar fractions.

The MN analysis revealed polyphenolic compounds, especially methoxylated flavonoids such as hispidulin and cirsiliol, as contributors to the observed antitubercular activity in *T. riparia* and *Z. zerumbet.* Commercial standards confirmed the predictive value of the network, validating the strategy’s potential to distinguish active from inactive metabolites, even among structurally similar compounds. In contrast, certain terpenoid-rich fractions displayed limited bioactivity, underscoring the importance of coupling chemical annotation with biological data. These findings highlight the potential of natural products in the research of new drug candidates. However, precise target identification and in vivo validation of the most promising compounds are required to confirm their mechanism of action and therapeutic relevance. Such studies will help to consolidate their translational potential and guide their integration into the TB drug discovery pipeline.

While some differences with previously published bioactivities were observed, likely due to differences in plant origin or extraction protocols, the MN-based strategy proved robust in deconvoluting complex mixtures and identifying both known and novel bioactive scaffolds. Future work involving targeted isolation and biological evaluation of key metabolites, particularly minor alkaloids from *I. suffruticosa* and minor terpenoids from *T. riparia* and *Z. zerumbet,* will further expand the chemical space of potential antitubercular agents. Overall, this integrative approach offers a scalable and efficient alternative to classical bio-guided isolation, with applications for natural product-based drug discovery in the fight against tuberculosis.

## Figures and Tables

**Figure 1 plants-14-03028-f001:**
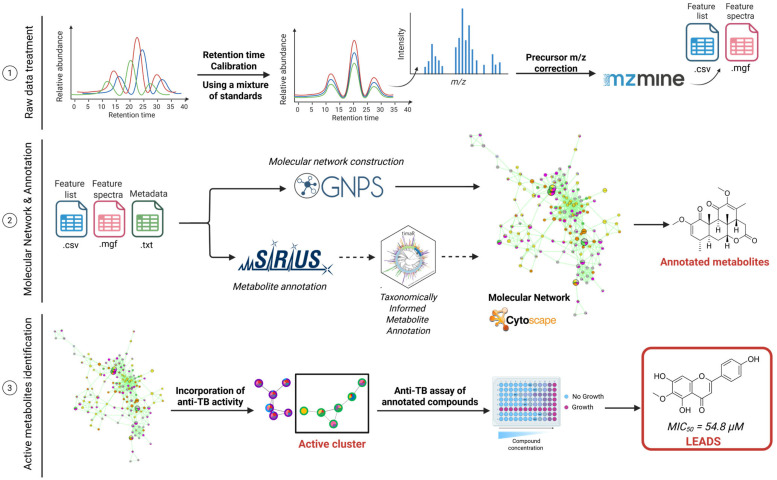
Schematic representation of the automated workflow used for the high-throughput annotation of bioactive metabolites from plant extracts. The process involves (1) raw data treatment, (2) molecular networking and annotation, and (3) active metabolites annotation. This pipeline enables reproducibility and rapid dereplication of candidate antitubercular leads.

**Figure 2 plants-14-03028-f002:**
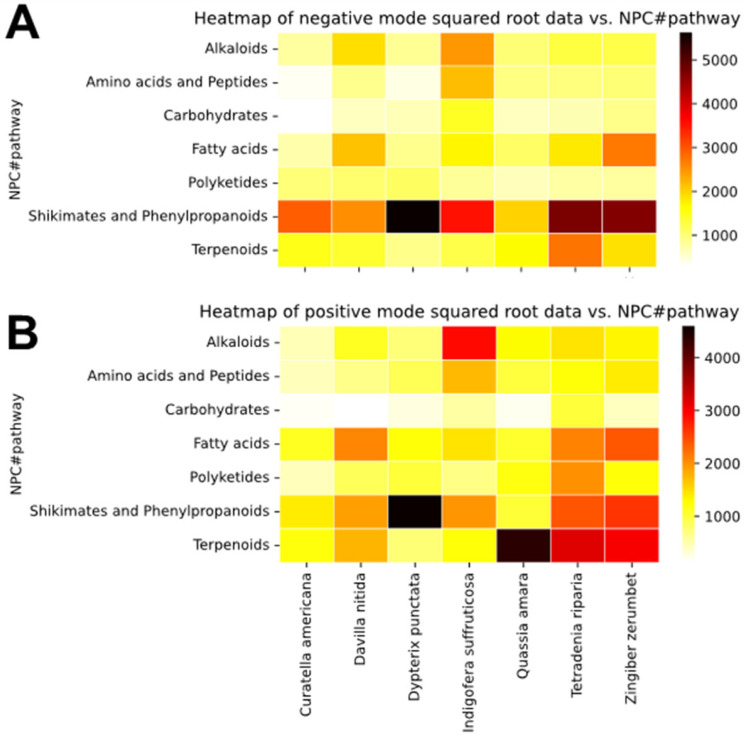
Heatmaps showing the relative metabolite composition across species in negative (**A**) and positive (**B**) modes, according to metabolic pathways. Heatmaps were generated based on SIRIUS predictions, following the Natural Products Classifier classification.

**Figure 3 plants-14-03028-f003:**
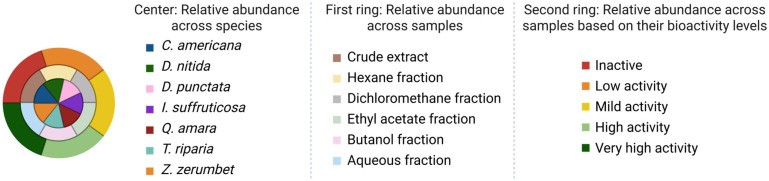
Nodes pie chart color code. Relative abundance is calculated based on peak area of the feature in each sample. For the second ring, representing relative abundance based on bioactivity levels, the corresponding levels are: MIC_50_ > 500 µg/mL = inactive; 250 < MIC_50_ < 500 µg/mL = low activity; 150 < MIC_50_ < 250 µg/mL = mild activity; 75 < MIC_50_ < 150 µg/mL = high activity; and MIC_50_ < 75 µg/mL = very high activity.

**Figure 4 plants-14-03028-f004:**
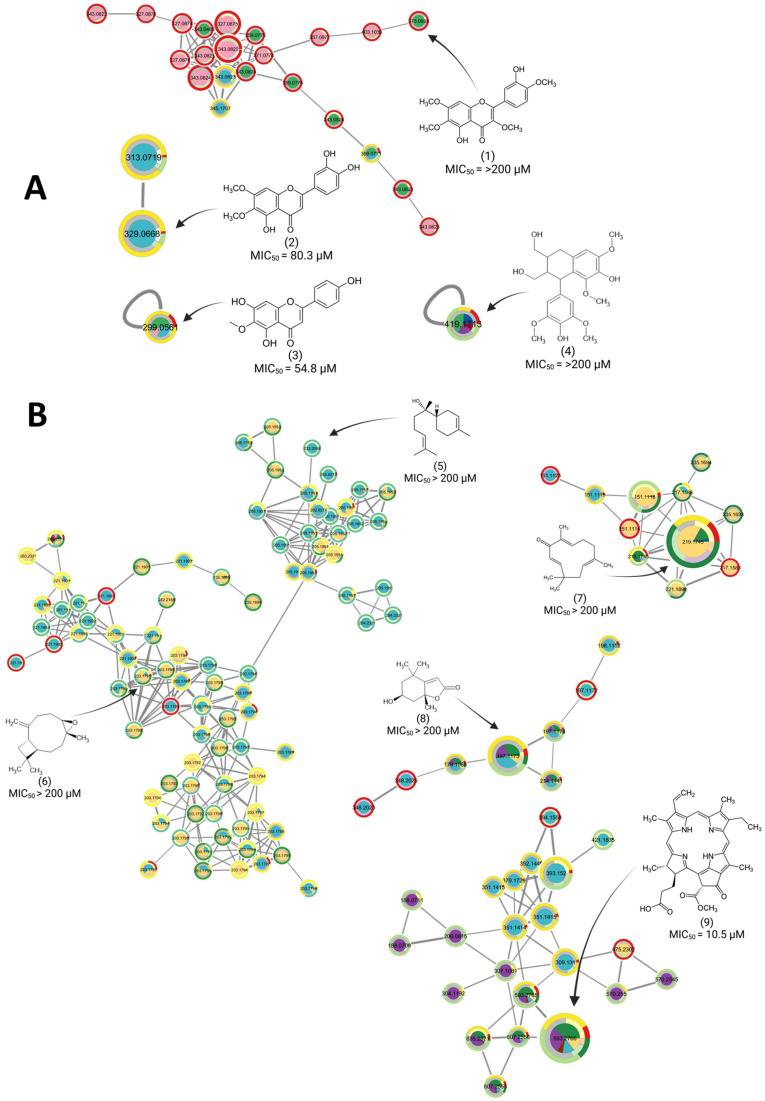
(**A**) Clusters, on the negative ionization MN, of compounds tested against Mtb H37Ra. (**B**) Clusters, on the positive ionization MN, of annotated compounds tested against Mtb H37Ra. All corresponding MIC can be found in Table 2. (1) Casticin; (2) cirsiliol; (3) hispidulin; (4) lyoniresinol, (5) α-bisabolol; (6) caryophyllene oxide; (7) zerumbone; (8) loliolid; (9) pheophorbide A.

**Figure 5 plants-14-03028-f005:**
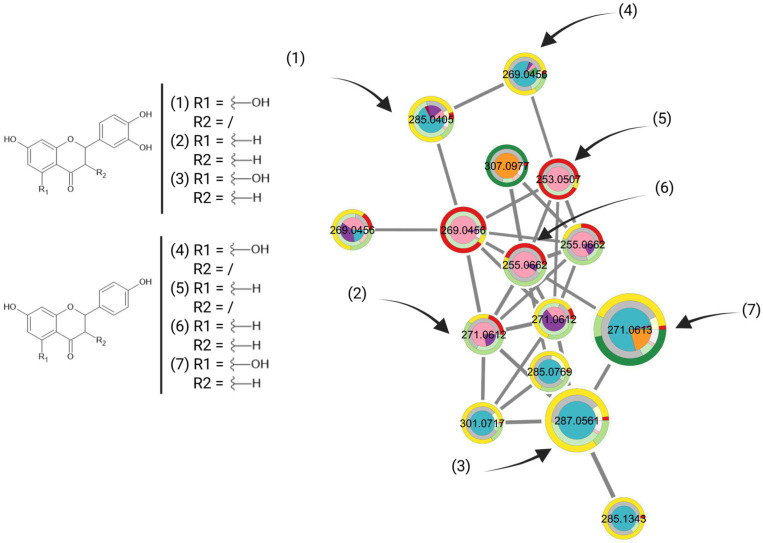
Cluster, on the negative ionization MN, of flavonoids and their previously reported MIC_90_s against *Mtb*. (1) Luteolin MIC (H37Rv) = 78.12 µg/mL; (2) 3′,4′,7-Trihydroxyflavanone MIC (H37Rv) = 100 µg/mL; (3) eriodictyol, no reported MIC; (4) apigenin MIC (H37Rv) = 70 µg/mL; (5) 4′,7-Dihydroxyflavone, no reported MIC; (6) liquitirigenin MIC (H37Ra) = 25 µg/mL; (7) naringenin MIC (H37Rv) ≤ 2.8 µg/mL [46].

**Table 1 plants-14-03028-t001:** Antitubercular activities of the most active plant extracts against *Mtb* H37Ra.

Sample Code	MIC50 [µg/mL]	MIC90 [µg/mL]
IS-L-E-HE	244 ± 67	>500
IS-L-E-DM	147 ± 39	>500
IS-L-D-EA	166 ± 69	>500
TR-L-E-HE	128 ± 32	>500
TR-L-E-DM	178 ± 66	425 ± 130
ZZ-L-E	32 ± 17	>500
ZZ-L-E-HE	211 ± 69	261 ± 96
ZZ-R-E-HE	130 ± 64	253 ± 166
ZZ-R-E-DM	74 ± 30	176 ± 39

All reported values are expressed as mean ± standard deviation (SD) of at least three independent assays performed in duplicate. Details for the code names of the samples are given in 4.2.3. IS = *Indigofera suffruticosa*; TR = *Tetradenia riparia*; ZZ = *Zingiber zerumbet*; L = leaves; R = rhizome; E = UA hydro-ethanolic extract; D = decoction extract; HE = hexane fraction; DM = dichloromethane fraction; EA = ethyl acetate fraction.

**Table 2 plants-14-03028-t002:** MIC_50_ and MIC_90_ (µM) of selected pure compounds against *Mtb* H37Ra.

Compound	MIC_50_ (µM)	MIC_90_ (µM)	Species Identified in
(1) Casticin	>200	>200	*D. nitida*
(2) Cirsiliol	80.3 ± 43	>200	*D. nitida*; *T. riparia*
(3) Hispidulin	54.8 ± 25.6	162 ± 75	*D. nitida*; *D. punctata*; *T. riparia*
(4) Lyoniresinol	>200	>200	*C. americana*; *D. nitida*; *I. suffruticosa*; *Q. amara*
(5) α-bisabolol	>200	>200	*T. riparia*
(6) Caryophyllene oxide	>200	>200	*T. riparia*; *Z. zerumbet*
(7) Zerumbone	>200	>200	*Z. zerumbet*
(8) Loliolide	>200	>200	*D. nitida*; *I. suffruticosa*; *Q. amara*; *T. riparia*; *Z. zerumbet*
(9) Pheophorbide A	10.5 ± 5.0	17.0 ± 7.6	*D. nitida*; *I. suffruticosa*; *Q. amara*; *T. riparia*; *Z. zerumbet*
Kanamycin *	3.3 ± 0.40	7.5 ± 1.7	
Ethambutol *	10.3 ± 4.0	17.3 ± 0.96	

* Positive control. All reported values are expressed as mean ± SD of at least three independent assays performed in duplicate. Structures of each compound are depicted in Figure 4.

**Table 3 plants-14-03028-t003:** Cytotoxic activity of 5 selected NPs against mammalian cells ***^a^***.

Compounds	BEAS-2B		IMR90		HepG2	
	CC_50_ [µM]	TI	CC_50_ [µM]	TI	CC_50_ [µM]	TI
(2) Cirsiliol	>250	Nd	107.6 ± 44.1	<0.54	>250	>1.25
(3) Hispidulin	>250	>1.5	145.9 ± 62.7	0.9	>250	>1.5
(5) α-bisabolol	>250	Nd	>250	Nd	>250	>1.25
(7) Zerumbone	149 ± 43.2	<0.74	79.8 ± 48.7	<0.4	>250	>1.25
(9) Pheophorbide A	4.9 ± 2.7	0.29	11.5 ± 10.5	0.68	>84.3	>4.9

***^a^*** Experiments were performed as described in the Material and Methods Section 4.6. CC_50_: Cytotoxic concentration leading to 50% cell death in vitro compared to the DMSO control. Data are expressed as mean ± SD (n = 3). TI: Therapeutic index calculated by dividing the CC_50_ by the MIC_90_ reached for the growth of *Mtb* H37Ra (see Table 2). Nd: Non-determined.

**Table 4 plants-14-03028-t004:** Plant collection data.

Taxon	Plant Part	Family	Location	Collect Date	Voucher Specimen Barcode
*Curatella americana*	Bark	Dilleniaceae	Savane des Mornes; Macouria Lat, lon: N 4°59′46.5′′, W 52°28′00.1′′	1 December 2022	CAY252733
*Davilla nitida*	Leaves	Dilleniaceae	Route de l’Est; Roura Lat, lon: N 4°42′07.4′′, W 52°23′03.3′′	19 December 2022	CAY252737
*Dipteryx punctata*	Bark	Leguminosae	Secteur Patawa; Montsinery Lat, lon: N 4°50′25.9′′, W 52°30′17.9′′	1 December 2022	CAY252738
*Indigofera suffruticosa*	Leaves	Leguminosae	Route de Lance; Sinnamary Lat, lon: N 5°22′08.7′′, W 52°54′11.5′′	22 March 2022	CAY252734
*Quassia amara*	Wood	Simaroubaceae	PK13 route de Macouria; Macouria Lat, lon: N 4°55′44.3′′, W 52°23′45.7′′	18 November 2024	CAY252742
*Tetradenia riparia*	Leaves	Lamiaceae	La Carapa; Macouria Lat, lon: N 4°54′47.9′′, W 52°26′59.7′′	14 March 2023	CAY252743
*Zingiber zerumbet*	Leaves and Rhizome	Zingiberaceae	Quartier belle terre; Macouria Lat, lon: N 4°55′28.4′′, W 52°23′21.1′′	1 December 2022	CAY252735 & CAY252736

## Data Availability

Raw data of LC-MS/MS analysis were deposited in the MassIVE Public dataset available at ftp://massive-ftp.ucsd.edu/v10/MSV000098589/ (accessed on 22 July 2025). The molecular networking job on GNPS can be found at https://gnps.ucsd.edu/ProteoSAFe/status.jsp?task=a2ed23823eed46fcb6b1081735588de9 (negative ionization mode job, accessed on 10 July 2025) and https://gnps.ucsd.edu/ProteoSAFe/status.jsp?task=8ee98816e7ff4b1aad3ede4355266b73 (positive ionization mode job, accessed on 10 July 2025). The exports obtained from MZmine, SIRIUS, TIMA-R, and Cytoscape networks were uploaded to Zenodo repository (http://dx.doi.org/10.5281/zenodo.16086397, accessed on 25 July 2025).

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
