# Peer review of "Bioactive Compounds Discovery from French Guiana Plant Extracts Through Antitubercular Screening and Molecular Networking"

_plants, 2025, doi:10.3390/plants14193028_

Round 1

Reviewer 1 Report (Previous Reviewer 2)

Comments and Suggestions for Authors

The manuscript submitted for review is aimed at searching for plant materials containing metabolites exhibiting anti-tuberculosis activity. To simplify the component composition of the studied extracts, the authors used solvents of varying polarity. This made it possible to identify fractions with the greatest anti-tuberculosis activity.

The identification of metabolites by HPLC-MS raises the greatest doubts. The manuscript overuses automated results processing. Given the specifics of the mass spectrometer used, authors are encouraged to provide the precursor ion isolation window. If a comparison with online MS/MS libraries was performed, a comparison with the experimental spectrum can be provided in the supplementary materials to demonstrate the absence of interfering isotopic peaks (if it is impossible to isolate monoisotopic peaks during fragmentation). It would also be helpful to provide chromatograms, for example, for the most promising fractions. 

The manuscript mentions that decoctions of the plants studied were also obtained. It would be helpful to provide a more detailed description of what was found in these decoctions, in comparison with ultrasonic extraction.

The authors also often talk about nine standard samples, but no information is provided in the experimental section about them.

Some results, for example the yield of extractive substances, have an excessive number of significant digits.

Author Response

We would like to really thank the Reviewers for their positive evaluations and constructive comments that will help us to improve the quality of our manuscript.

You will find below our point-by-point response to the new comments. All modifications have been highlighted in yellow as clearly indicated in the marked revised version of the manuscript.

The manuscript submitted for review is aimed at searching for plant materials containing metabolites exhibiting anti-tuberculosis activity. To simplify the component composition of the studied extracts, the authors used solvents of varying polarity. This made it possible to identify fractions with the greatest anti-tuberculosis activity.

Question 1. The identification of metabolites by HPLC-MS raises the greatest doubts. The manuscript overuses automated results processing.

Reply 1. We are sorry that you had impression of automated results processing. Actually, we wanted to show, that all these automatic tools allow go faster in the identification, however all the results presented in this paper were manually checked and compared with experimental and in absence of experimental with in silico data. That’s why we used the confidence level in the metabolite annotation.

Question 2. Given the specifics of the mass spectrometer used, authors are encouraged to provide the precursor ion isolation window.

Reply 2. As requested, the precursor isolation window was added to the Materials and Methods section (see lines 748).

Question 3. If a comparison with online MS/MS libraries was performed, a comparison with the experimental spectrum can be provided in the supplementary materials to demonstrate the absence of interfering isotopic peaks (if it is impossible to isolate monoisotopic peaks during fragmentation).

Reply 3. We thank the reviewer for this suggestion. For all annotated compounds, experimental MS/MS spectra acquired were compared to online MS/MS libraries spectra (GNPS, Massbank, or spectra shared in articles), and the obtained experimental fragment ions are already reported in the annotation table (Table S3). In addition, the GNPS jobs are publicly available, allowing independent verification of the spectral matches and ensuring transparency of the annotation process. If annotation was not confirmed through GNPS, the corresponding reference was added in Table S3.

As mentioned in the Data Availability Statement section, a visual comparison of experimental MS/MS spectra realized on GNPS platform is available through the molecular network links in the View All Library Hits part and View mirror match link. The molecular networking job on GNPS can be found at https://gnps.ucsd.edu/ProteoSAFe/status.jsp?task=a2ed23823eed46fcb6b1081735588de9 (negative ionization mode job) and https://gnps.ucsd.edu/ProteoSAFe/status.jsp?task=8ee98816e7ff4b1aad3ede4355266b73 (positive ionization mode job).

As an example we present here a MS/MS spectra mirror match of Formononetin:

We appreciate the concerns of the Reviewer regarding the isolation window. It is true that the isolation window in Bruker instruments is wide with 2.5 Da windows from each side of precursor ion to ensure a good sensibility for MS2 spectra. Actually, the fragmentation spectra generated by M+1 ion rests lower than 10% and doesn’t strongly impact the MS2 spectra interpretation. All the annotations were curated manually after all the automatic algorithms to ensure their correct identification. We would also like to cite the work of Nothias et al. (10.1038/s41592-020-0933-6) and Schimd et al. (10.1038/s41587-023-01690-2), the reference papers in the field of feature based molecular networks and MZmine software, where the authors used the timsTOF instrument from Bruker which follows the same isolation windows. In the study of Stancliffe et al. (10.1038/s41592-021-01195-3), the authors compared different MS2 spectra using 1,3 and 5 Da width. The isolation windows of 1-3 Da are considered as narrow windows. The authors showed that there is not much difference of dot product similarity score observed between MS2 spectra obtained using 1, 3 and 5 Da isolation windows before deconvolution using the software developed by authors. The authors claimed that even 1 Da isolation window can generate contaminated MS2 spectra.

Question 4. It would also be helpful to provide chromatograms, for example, for the most promising fractions. 

Reply 4. We added 3 BPC chromatograms for the most promising fractions in positive and negative ionization modes. (Supplementary materials Figure S2 and S3; line 262-263 in the main text).

Question 5. The manuscript mentions that decoctions of the plants studied were also obtained. It would be helpful to provide a more detailed description of what was found in these decoctions, in comparison with ultrasonic extraction.

Reply 5. Here, the extracts and fractions obtained by decoction contained mainly polar compounds which showed no or very weak antibacterial activity against Mycobacterium tuberculosis. That’s why we didn’t focus on decoction comparison with UAE in this study.  This point is now clearly stated in the manuscript (see lines 477).

Question 6. The authors also often talk about nine standard samples, but no information is provided in the experimental section about them.

Reply 6. We apologize for this inappropriate wording. The “nine standard samples” are referred to the nine commercial molecules tested against Mtb. This point has been clearly stated in the Material and Methods section (see lines 755-757), and the wording corrected all along the text.

Question 7. Some results, for example the yield of extractive substances, have an excessive number of significant digits.

Reply 7. The number of significant digits after dot was decreased to 1 in Table S1 in the Supplementary Materials.

Reviewer 2 Report (New Reviewer)

Comments and Suggestions for Authors

The manuscript represent an interesting contribution to the field of bioactivity screening of plant species. I have to admit that it is not easy for a non-specialist to follow the leads of the work, in particular not the pie-chart (fig 3) and the active use of it. This could perhaps been explained better.

Parts of the text have been marked yellow and green. I guess this was not intended to be part of the submission(?) :)

The organization of the manuscript doesn't always fit with the paragraph headings. E.g. lines 126-144 of the result section gives a method description. In general, the text should be improved with respect to conciceness, and the different elements of the text should be arranged to the specific sections respectively (methods, results, discussion). Much space is used to explain MN under the result-section. This could perhaps been merged with what is said in the introduction section?

Lines 170-186: Again, there are few results in this text, rather description of the methods. The heatmap itself though, is a result.

Section 2.3: Again, results appear late, from line 224. 

Line 224-230: Don't repeat what is already said in table 1.

Line 241: "good results" is not a proper term. Rather "antibacterial activity"? :)

Lines 246-249: Move to discussion section.

Section 2.4: Most of what is described until line 336 are not results rather method description.

Discussion:

  • As MN is rather unfamiliar to me and I guess to other readers with phytochemical background, I would like to see a discussion of dis-/advantages of the present methodology compared to bioactivity guided fractionation -methodologies.
  • An obvious point of discussion is connected to the chosen extraction protocol. The highest activity seems to be found with lipophilic extracts. This means that crude extracts from e.g. dichloromethane might give even higher activity than crude extracts based on 70% ethanol.
  • Further, what about monoterpenoids? As you point out, there have been reports from T.riparia that essential oil exhibit activity towards tuberculosis.

lines 610-620: This is kind of a summary. Merge it with the conclusion section.

Author Response

We would like to really thank the Reviewers for their positive evaluations and constructive comments that will help us to improve the quality of our manuscript.

You will find below our point-by-point response to the new comments. All modifications have been highlighted in yellow as clearly indicated in the marked revised version of the manuscript.

Question 1. The manuscript represents an interesting contribution to the field of bioactivity screening of plant species. I have to admit that it is not easy for a non-specialist to follow the leads of the work, in particular not the pie-chart (fig 3) and the active use of it. This could perhaps be explained better.

Reply 1. We thank the reviewer for pointing out the need for clarification. In the molecular networks, each node represents a detected feature (putative metabolite), and the associated pie chart encodes three layers of information:

  • Center (core of the pie chart): This part reflects in which plant species the feature was detected, and its relative abundance across species. For example, if a metabolite is present mainly in riparia, the central color will correspond largely to that species.
  • First ring (middle ring): This layer shows the distribution of the same feature across the different extracts or fractions obtained for that species (n-hexane, dichloromethane, ethyl acetate, n-butanol, aqueous). It therefore allows visualization of the polarity of the metabolite.
  • Outer ring (external ring): This layer links chemistry to biology by indicating the relative abundance of the feature in fractions for which antitubercular activity was measured. The activity of each fraction is color-coded according to its MIC result, making it possible to highlight which features are enriched in active fractions.

By layering these three dimensions on a single node, the molecular network allows simultaneous visualization of taxonomic origin, extraction/fraction distribution, and biological activity, which greatly facilitates the prioritization of metabolites for further study.

This explanation was added in section 2.4., line 273-282.

Question 2. Parts of the text have been marked yellow and green. I guess this was not intended to be part of the submission(?) :)

Reply 2. The yellow and green marked parts are corresponding to changes between the initial submission and the revised version of the manuscript. For a better clarity, all these modifications have been cleaned, and all changes in actual review have been highlighted in yellow.

Question 3. The organization of the manuscript doesn't always fit with the paragraph headings. E.g. lines 126-144 of the result section gives a method description. In general, the text should be improved with respect to conciseness, and the different elements of the text should be arranged to the specific sections respectively (methods, results, discussion). Much space is used to explain MN under the result-section. This could perhaps be merged with what is said in the introduction section?

Reply 3. We thank the reviewer for this constructive comment. The manuscript was carefully reorganized to ensure that methodological details now appear exclusively in the Materials and Methods section, while the Results section is limited to experimental findings. Redundant explanations of the molecular networking approach that previously appeared in the results have been merged into the Introduction for conciseness and to avoid repetition.

Question 4. Lines 170-186: Again, there are few results in this text, rather description of the methods. The heatmap itself though, is a result.

Reply 4. The heatmaps were built based on a script that was written in the lab. We therefore kept in the Results section the explanation necessary for a better comprehension of the results following. The part regarding context of the function of SIRIUS was moved to Material&Methods, section 4.4.4.

Question 5. Section 2.3: Again, results appear late, from line 224. 

Reply 5. The phrase was kept for the contextualization to remind the experiment.

Question 6. Line 224-230: Don't repeat what is already said in table 1. Line 241: "good results" is not a proper term. Rather "antibacterial activity"? :)

Lines 246-249: Move to discussion section.

Reply 6. We kindly thank the reviewer for these remarks. The corresponding changes have been made in the main text. The literature comparison for T. riparia was moved to Discussion

Question 7. Section 2.4: Most of what is described until line 336 are not results rather method description.

Reply 7. Following Question 3, the description was moved to the Introduction, and shorten to stay concise.

Discussion:

Question 8. As MN is rather unfamiliar to me and I guess to other readers with phytochemical background, I would like to see a discussion of dis-/advantages of the present methodology compared to bioactivity guided fractionation -methodologies.

Reply 8. We thank the reviewer for this suggestion. A paragraph has been added to the Discussion (line 489-501) to directly compare the advantages and limitations of the molecular networking (MN) approach with conventional bioactivity-guided fractionation. We now emphasize that MN accelerates prioritization of bioactive candidates, helps discriminate between active and inactive metabolites within ubiquitous classes, and reduces the risk of repeatedly isolating known compounds. On the other hand, MN relies on in silico annotations and predictions, which require experimental confirmation through isolation and full structural elucidation. This discussion highlights that MN and bioactivity-guided fractionation should be considered complementary rather than mutually exclusive strategies.

Question 9. An obvious point of discussion is connected to the chosen extraction protocol. The highest activity seems to be found with lipophilic extracts. This means that crude extracts from e.g. dichloromethane might give even higher activity than crude extracts based on 70% ethanol.

Reply 9. We appreciate this observation. Our extraction protocol, based primarily on 70% ethanol (UAE) followed by partitioning, was designed to maximize metabolite diversity rather than to optimize for activity alone. While lipophilic fractions indeed showed the highest antitubercular activity, we note that direct extraction with highly non-polar solvents such as dichloromethane could yield in lower concentration of bioactive compounds because of higher molecular diversity. Also, this would have limited our ability to capture polar compounds of interest. The chosen approach therefore provided a balanced coverage of both polar and non-polar metabolites, while subsequent fractionation allowed us to focus on the most active lipophilic components. The direct extraction by non-polar solvents can be useful for the future purification of compounds of interest in order to increase their yield.

Question 10. Further, what about monoterpenoids? As you point out, there have been reports from T. riparia that essential oil exhibit activity towards tuberculosis.

Reply 10. Thank you for raising this point. Essential oils of T. riparia are indeed reported to display antitubercular activity, with compounds such as 6,7-dehydroroyleanone implicated as active diterpenoids. In our study, however, the extraction protocols were not optimized for the volatile fraction, and LC-MS/MS analysis is less suited to detect highly volatile monoterpenoids. Moreover, the described active compound 6,7-dehydroroyleanone was detected in the active fraction of T. riparia, with relatively low detection in LC-MS, as mentioned in the text (line 417).

Question 11. lines 610-620: This is kind of a summary. Merge it with the conclusion section.

Reply 11. We agree, thank you for the comment. The last paragraph of the Discussion was removed as it was redundant with the conclusion.

Reviewer 3 Report (New Reviewer)

Comments and Suggestions for Authors

Dear Authors,

Thank you for submitting your manuscript, "Bioactive compounds discovery from French Guiana plant extracts through antitubercular screening and Molecular Networking," to our journal. This is a very well-conducted study that addresses a crucial global health challenge: tuberculosis (TB) drug discovery. While the manuscript presents valuable data and a strong methodological foundation, I believe further refinements and experimental enhancements are necessary. My decision is Major Revisions, and I hope the following detailed suggestions will help you strengthen your manuscript significantly.

1. It is crucial to clearly articulate the unique contribution and novelty of your findings beyond just the methodological approach. While your integrated molecular networking (MN) approach is excellent, many of the identified active compounds (e.g., Hispidulin, Cirsiliol, Castin) are relatively well-known natural products. Please explicitly emphasize what makes the discovery of these specific compounds from these specific sources, or any potentially novel scaffolds/derivatives, particularly impactful. Did any compounds show exceptional potency, superior selectivity, or a unique mechanism of action compared to existing anti-TB drugs or other reported natural products? A more direct comparison to first-line anti-TB drugs (e.g., isoniazid, rifampicin) in your assays would help contextualise the observed MIC values and demonstrate relevance to clinical treatment.

2. Please consider providing more depth regarding the biological activity. While full mechanistic studies may be beyond the scope of this paper, some preliminary discussion or in silico predictions of targets for your most potent and selective compounds would be highly valuable. This would move the paper beyond pure screening towards a deeper biological understanding.

3. Your stated objective is "natural product-based drug development against TB". While this study is an excellent first step, the journal would expect a stronger narrative towards identifying truly promising lead candidates. Based on your data, which compounds would you strongly advocate for further preclinical development and why? A more explicit discussion of druggability criteria, predicted ADME properties (even if in silico), or potential for chemical modification to improve efficacy or reduce toxicity would strengthen this aspect significantly.

4. The discussion section is generally well-structured, but it could be enhanced by more directly addressing the broader implications of your findings for TB drug discovery, connecting them more explicitly to the current challenges in the TB drug pipeline, and outlining clear, impactful future directions. Importantly, explicitly discuss the limitations of the current study (e.g., lack of in vivo data, precise target identification, and comprehensive SAR exploration beyond clustering) to maintain scientific rigor and transparency.

Author Response

We would like to really thank the Reviewers for their positive evaluations and constructive comments that will help us to improve the quality of our manuscript.

You will find below our point-by-point response to the new comments. All modifications have been highlighted in yellow as clearly indicated in the marked revised version of the manuscript.

Question 1. It is crucial to clearly articulate the unique contribution and novelty of your findings beyond just the methodological approach. While your integrated molecular networking (MN) approach is excellent, many of the identified active compounds (e.g., Hispidulin, Cirsiliol, Castin) are relatively well-known natural products. Please explicitly emphasize what makes the discovery of these specific compounds from these specific sources, or any potentially novel scaffolds/derivatives, particularly impactful. Did any compounds show exceptional potency, superior selectivity, or a unique mechanism of action compared to existing anti-TB drugs or other reported natural products? A more direct comparison to first-line anti-TB drugs (e.g., isoniazid, rifampicin) in your assays would help contextualise the observed MIC values and demonstrate relevance to clinical treatment.

Reply 1. Our research work focuses on the identification of bioactive metabolites using automated tools. Our main goal was to show that automatic processing can provide faster and efficient results, but manual curation is still necessary to ensure the correct identification of compounds. The novelty of this work resides on the study of phytochemical composition of plant species, which are almost not known. We are well aware that significant optimization work would be required for the tested compounds to become candidates for future drugs. However, some molecules, such as Hispidulin, shows very interesting profile in terms of toxicity/antibacterial activity ratio. In this special case, conducting a SAR study would indeed lead to significant improvements. However, this would represent a considerable amount of work, that is beyond the scope of the current article, and could be considered a study in its own right. Regarding comparison with known anti-TB drug, the MICs of ethambutol and kanamycin have been added in Table 2.

Question 2. Please consider providing more depth regarding the biological activity. While full mechanistic studies may be beyond the scope of this paper, some preliminary discussion or in silico predictions of targets for your most potent and selective compounds would be highly valuable. This would move the paper beyond pure screening towards a deeper biological understanding.

Reply 2. We thank reviewer for this suggestion. We have added a few sentences to discuss the potential targets of the most promising compound (see lines 621-624).

Question 3. Your stated objective is "natural product-based drug development against TB". While this study is an excellent first step, the journal would expect a stronger narrative towards identifying truly promising lead candidates. Based on your data, which compounds would you strongly advocate for further preclinical development and why? A more explicit discussion of draggability criteria, predicted ADME properties (even if in silico), or potential for chemical modification to improve efficacy or reduce toxicity would strengthen this aspect significantly.

Reply 3. We thank reviewer for this suggestion. We have added in the Discussion section a new paragraph dealing with ADME properties for the most promising compound hispidulin (see lines 625-632).

Question 4. The discussion section is generally well-structured, but it could be enhanced by more directly addressing the broader implications of your findings for TB drug discovery, connecting them more explicitly to the current challenges in the TB drug pipeline, and outlining clear, impactful future directions. Importantly, explicitly discuss the limitations of the current study (e.g., lack of in vivo data, precise target identification, and comprehensive SAR exploration beyond clustering) to maintain scientific rigor and transparency.

Reply 4. We thank the reviewer for this suggestion. We have added a new paragraph in the Conclusion section dealing with these issues to maintain scientific rigor and transparency (see lines 907-911).

Round 2

Reviewer 1 Report (Previous Reviewer 2)

Comments and Suggestions for Authors

The authors made some corrections to the manuscript. Furthermore, in their responses to comments, they were able to address some concerns and convince the reviewer of the validity of their results.

Reviewer 2 Report (New Reviewer)

Comments and Suggestions for Authors

This revised version has taken into account some of my considerations, and I recommend this manuscript for publication in Plants.

Reviewer 3 Report (New Reviewer)

Comments and Suggestions for Authors

Dear Authors,

Thank you for your careful and thorough revisions.
All suggestions have been addressed in this revision. I have no further comments and support acceptance.

This manuscript is a resubmission of an earlier submission. The following is a list of the peer review reports and author responses from that submission.

Round 1

Reviewer 1 Report

Comments and Suggestions for Authors

This study represents a commendable integration of ethnopharmacological leads, metabolomics, molecular networking (MN), and bioactivity-guided screening to identify antitubercular compounds from plants native to French Guiana. The researchers' use of tools such as SIRIUS, GNPS, TIMA-R, and LC-MS/MS is well-justified and aligned with current best practices in natural product dereplication.

These are observed gaps that I saw during reading this paper:

Dereplication and in silico annotation are strong points, but the lack of experimental isolation and full structural elucidation (NMR, X-ray, etc.) for novel candidates weakens claims of novel discovery. Prioritize purification and comprehensive spectroscopic characterization of the most active fractions (especially from I. suffruticosa and T. riparia) to confirm structures and bioactivities.

The predicted activity based on MN is only partly validated through MIC and CC50 assays. Some compounds with promising annotations (e.g., boronolide) were not tested. Expand bioactivity validation of annotated compounds with borderline predictions. Include quantitative structure-activity relationships (QSAR) or SAR discussions where feasible.

MN predictions for terpenoids and less ubiquitous metabolites showed poor alignment with experimental data (i.e., false positives such as α-bisabolol and zerumbone). Incorporate a confidence-scoring system in MN cluster prioritization and include cross-validation with known standards more systematically.

Cytotoxicity was tested on a limited set of mammalian cells (BEAS-2B, IMR90) and did not include hepatocytes or PBMCs which are relevant for first-pass metabolism and systemic toxicity. Expand cytotoxic profiling to include hepatocytes and broader immune cell lines to better estimate the therapeutic window of promising compounds.

The study mentions ethnopharmacological knowledge, but the connection between traditional use and selection rationale for the tested plants is underdeveloped. Provide more robust ethnobotanical context (e.g., traditional medicinal uses, prior antimicrobial reports) for each plant to strengthen the rationale for selection.

Standard deviations in bioassay tables are sometimes large, and the number of replicates isn't always clear. Ensure all bioactivity results include number of replicates and p-values for significant comparisons. Consider multivariate analysis (e.g., PCA) to complement heatmap visuals.

Reviewer 2 Report

Comments and Suggestions for Authors

The manuscript is devoted to an important area: the search for new sources of biologically active components that have high potential in the fight against various dangerous diseases, including tuberculosis. After carefully reading this article, many doubts and questions remain. The main ones are:

The text contains many links to Supplementary Materials, but they are not in the system. This limits the perception of some information.

Experimental Methodology: It is necessary to clarify why the authors used the decoction method in addition to ultrasonic extraction. From the text, this choice is not obvious. 

Line 326. The authors talk about nine components that they tested in their work. Why aren't these experiments mentioned in the methodology section?

Identification of metabolites. The most controversial part. The authors use liquid chromatography with high-resolution mass-spectrometry detection for this purpose. The work uses an ultra-fast separation mode (2.1 × 100 mm, 1.8 µm column, flow 0.8 ml/min). But for detection, a scanning speed of only 4 Hz is used. There are doubts that this is sufficient, given the continuous recording of tandem mass spectra. This mass spectrometer has a feature - the precursor ion isolation window cannot be less than 2 Da. This implies simultaneous entry into the collision cell of the target ion and isotope. This greatly complicates mass spectra. Judging by the description, the search and identification of metabolites in extracts is carried out completely automatically using software. There are serious doubts about the correctness of the establishment of the component composition in the isolated extracts.

There is a feeling that the authors overloaded the work by including 7 plant species at once. It would have been better to focus in detail on one species or one family, conduct a reliable identification, compare with the data available in the literature, including the composition of key metabolites and anti-tuberculosis activity. It is not clear now what is the novelty of this work? What is the contribution of the study to the search for anti-tuberculosis biologically active compounds?

The manuscript requires revision.